# A Stretchable Expanded Polytetrafluorethylene-Silicone Elastomer Composite Electret for Wearable Sensor

**DOI:** 10.3390/nano13010158

**Published:** 2022-12-29

**Authors:** Jianbo Tan, Kaikai Chen, Jinzhan Cheng, Zhaoqin Song, Jiahui Zhang, Shaodi Zheng, Zisheng Xu, Shiju E

**Affiliations:** 1Key Laboratory of Urban Rail Transit Intelligent Operation and Maintenance Technology & Equipment of Zhejiang Province, College of Engineering, Zhejiang Normal University, Jinhua 321004, China; 2Jinhua Intelligent Manufacturing Research Institute, Jinhua 321004, China

**Keywords:** stretchable electrets, high surface potential, wearable sensor, electrostatic induction effect, longevity, stretchability

## Abstract

Soaring developments in wearable electronics raise an urgent need for stretchable electrets. However, achieving soft electrets simultaneously possessing excellent stretchability, longevity, and high charge density is still challenging. Herein, a facile approach is proposed to prepare an all-polymer hybrid composite electret based on the coupling of elastomer and ePTFE membrane. The composite electrets are fabricated via a facile casting and thermal curing process. The obtained soft composite electrets exhibit constantly high surface potential (−0.38 kV) over a long time (30 days), large strain (450%), low hysteresis, and excellent durability (15,000 cycles). To demonstrate the applications, the stretchable electret is utilized to assemble a self-powered flexible sensor based on the electrostatic induction effect for the monitoring of human activities. Additionally, output signals in the pressure mode almost two orders of magnitude larger than those in the strain mode are observed and the sensing mechanism in each mode is investigated.

## 1. Introduction

Electrets, a class of dielectric materials that can trap charges or dipoles for a long time, are ubiquitous in many daily applications, including microphones, air filters, energy harvesters, and radiation sensors [1,2,3,4,5,6,7]. To date, inorganic electrets (silicon dioxide [8,9], barium titanate [10], lead zirconate titanate [11], etc.) or hard polymeric electrets (polytetrafluoroethylene (PTFE) [12,13], polypropylene [14], polyvinylidene fluoride [15], etc.) exhibit high charge density and charge life but lack stretchability. Soft dielectrics such as elastomers bear sufficient stretchability but suffer short-lived charge [6]. Traditional electrets are difficult to meet the requirements for the rapidly developing flexible wearable devices, the stretchable and longevous electrets are intensively desirable and have been actively pursued [15,16,17,18,19]. Unfortunately, stretchability and longevity are usually mutually exclusive in the existing electrets. The contradiction between stretchability and longevity is becoming a stumbling block to meeting the requirements for the practical application of electrets in emerging soft technologies.

To resolve this issue, tremendous efforts have been devoted to tailoring the trade-off between stretchability and longevity as much as possible on the premise of high charge density in the development of stretchable electrets. The introduction of pores in hard electrets is an alternative route to make soft electrets [20,21]. Wang et al. reported a porous polymeric electret material based on PTFE, which presented excellent softness with high charge storage [3]. Moreover, the induced charge responds more significantly to external forces for the reduced stiffness. Nevertheless, the porous inorganics and plastics have relatively low stretches and easily collapsed for the lack ductility of electrets and the air space in the pores [20]. The flexible thin sheets are another candidate for preparing flexible electrets. Ma et al. [2] fabricated a stretchable wave-shaped fluorinated ethylene propylene electret film via a template patterning route. Although both stretchability and longevity were achieved in this electret film, the strain and stretch direction also restrict its applications. A straightforward approach has been proposed to improve the longevity of elastomer electrets, which is to introduce the inorganic nanoparticles or hard plastic nanoparticles, e.g., silica nanoparticles [9] and PTFE nanoparticles [22], into the elastomer matrix. Suo et al. [9] reported a stretchable electret based on silica nanoparticles filled polydimethylsiloxane, which provided a charge density of 4 × 10^−5^ C m^−2^ and a lifetime beyond 60 days. The nanoparticle-elastomer composites reached outstanding stretchability and longevity but low charge density. Other approaches are also actively considered, e.g., the use of block copolymers [5,16] and functional molecular liquids [23]. According to the research mentioned above, it is still a huge challenge to achieve a long-lived stretchable electret with high charge density.

Herein, we demonstrate a feasible approach to resolve the contradiction between longevity and stretchability on the premise of achieving high charge density for the electret. Namely, a thermally polarized ePTFE membrane consisting of interconnected nanopores and nanofibrous PTFE, is introduced into Ecoflex elastic matrix by a facile doctor-blade casting process. The resultant soft composite electret exhibits a constantly high surface potential (−0.38 kV) over a long time (30 days), large strain (450%), low tensile strength (0.8 MPa), and excellent durability (15,000 cycles). Additionally, a self-powered dual-modal flexible sensor was assembled based on the stretchable electret, and the sensing characteristics and mechanisms of this sensor in each sensing mode are demonstrated. Furthermore, owing to the excellent durability, water resistance, and skin-conformal of as-fabricated sensor, it is utilized to detect various real-time human physiology activities, including joint movements and various external force stimuli.

## 2. Experimental Section

*Fabrication of the stretchable electrets:* The ePTFE membrane (thickness of 50 μm; average pore size of 1 μm; Changzhou Jinchun Environmental Protection Technology Co., Ltd.) was charged in air condition with the assistance of metal grid, and the metal grid was 8 cm away from the ePTFE membrane (Figure 1a). The Ecofelx 00-30 (Smooth-On Inc.) solution part A and part B were mixed in a 1:1 ratio by a mixer. Then, the elastomer solution was cast onto the charged ePTFE film and flattened by a doctor blade on a glass plate. Finally, the Ecoflex precursor was solidified at room temperature for 4 h and post cured at 80 °C for 2 h to obtain the composite electret.

*Fabrication of the composite electret based sensor:* Firstly, the hydrogel electrode was synthesized following the procedure reported by Hyunwoo Yuk et al. [24]. The polyacrylamide hydrogel was synthesized by mixing 10 mL of degassed precursor solution (23 wt% acrylamide, 5 wt% NaCl, 0.051 wt% N,N-methylenebisacrylamide and 0.043 wt% Irgacure 2959). Subsequently, the precursor solution was coated on the surface of composite electret, and the gel was crosslinked by ultraviolet light irradiation for 2 h. Eventually, the hydrogel electrode and composite electret were sealed by Ecoflex to obtain the resultant sensor.

*Characterization:* The samples were brittle fractured in liquid nitrogen to exposure the cross section, on which a thin god layer was sputtered. The scanning electron microscopy (SEM, SU 8020) with an accelerate voltage of 5 kV was used to observe the morphology of cross-section. The mechanical properties of the composite electret were determined using a tensile test by an electronic universal material testing machine (Zwick, German). The surface potential was mapped by a high-speed electrometer (Model347, Trek, America). Light transmittance was measured by UV-Vis spectrophotometry (Agilent, Cary-60) in the range of 300–800 nm. The output signals of the stretchable electret and as-fabricated sensor were measured with a programmable electrometer (Keithley 6514). Contact angles were measured using a SL250 (Kono, US) apparatus at ambient temperature.

## 3. Results and Discussion

The ePTFE porous membrane and Ecoflex are selected as charge storing element and elastic matrix for stretchable electrets, respectively. The fabrication of stretchable electret is accomplished as schematically illustrated in Figure 1a. Firstly, the ePTFE porous membrane is polarized through a corona polarization treatment. A stable surface potential up to ≈−0.71 kV is obtained on the surface of polarized ePTFE film (Appendix A). Secondly, the charged membrane is coated with Ecoflex oligomer and flattened with a doctor blade. Owing to the interconnected porous network of ePTFE membrane (Appendix A) and good wettability of Ecoflex oligomer on the ePTFE membrane (Appendix A), the precursors can be thoroughly infiltrated into the ePTFE porous membrane. Finally, a thermal treatment is performed to accelerate the hydrosilylation addition reaction between the silicon precursors to crosslinking the silicone rubber, and the stretchable composite electret is achieved. The cross-sectional morphology of the composite is displayed, and no perceptible sign of pores or phase separation is observed (Figure 1b). Meanwhile, the Si elements are homogeneously distributed in the whole cross section as identified by the EDX mapping (Figure 1b), confirming complete penetration of Ecoflex precursor in porous film and an outstanding interface interaction between ePTFE film and Ecoflex matrix, which is profitable to the mechanical reliability and stability of composite electret. As seen from Fourier transform infrared spectroscopy (Figure 1c), the characteristic transmission peaks of Eptfe/Ecoflex composite are the same as those of ePTFE and Ecoflex, i.e., no new absorption peak is observed, indicating that no fresh chemical bond formed between ePTFE and Ecoflex elastomer in the course of the cross-linking process. In addition, because of the cellular-free and uniform structure, light scattering can be eliminated to enhance the transmittance of composite. As shown in Figure 1d, the composite electret possesses a transparency approximately 5–20% in the visible light range, and an intuitive comparison is demonstrated.

To assess the charge storage capacity of composite electret, the surface potential is measured and presented in Figure 1e. It is found to decay from an initial ≈−0.6 kV and then ultimately stabilize at ≈−0.38 kV over 30 days, showing a much longer charge storage time than that of Ecoflex elastomer (~20 h) (Appendix A). Theoretically, the charge can be maintained for hundreds of years [25], which ensures sustainable active status for the composite electret in practical applications. Generally, the robust mechanical property of composite electrets is extremely desirable for their practical applications. The composite electret, which can be rolled up, folded, twisted, and crumpled, exhibits outstanding flexibility (Appendix A). Compared with the pristine ePTFE membrane (Appendix A), the composite electret exhibits significantly enhanced elongation at break and reduced tensile stress, reaching 450% and 0.8 MPa, respectively (Figure 2f). Meanwhile, following an initial loading cycle, the composite electret exhibits very low mechanical hysteresis for 100 cycles of 200% strain (Figure 1g), indicating a marked improvement of elasticity in comparison with pure ePTFE film (Appendix A). The combination of Ecoflex matrix and ePTFE gives rise to the improved comprehensive mechanical properties of the composite electret because of the excellent stretchability, low elastic modulus, and low hysteresis of Ecoflex elastomer (Appendix A). In comparison with the recently reported research [5,6,9,17,20,22,26], the ePTFE/Ecoflex composite electret exhibits superior performance in terms of stretchability and longevity. Therefore, a composite electret with longevity, high charge density, and robust comprehensive mechanical properties is achieved, which can be potentially applicable to constructing self-powered flexible electronic devices.

Thanks to its superior stretchability, longevity, and high charge density, the composite electret is combined with a compliant hydrogel electrode and sealed by Ecoflex to fabricate a flexible single electrode sensor capable of perceiving and monitoring the external stimuli through the electrostatic induction. When a tensile strain is applied, the distance between electret film and electrode decreases with the deformation of sensor, and an increasing charge on the electrode is induced by the charged electret. In turn, the increasing gap and declining charge are evoked during the removal of external strain (Figure 2a). A simplified basic physical model of the stretchable composite electret is given in Appendix A and a more detailed elaboration of the sensing mechanism is presented in Appendix A. In Figure 2b, the influence of frequency on the output performance of this sensor is manifested. It can be found that the variation in output current of the sensor is in sync with the frequency of applied strain from 1 to 5 Hz, implying that the sensor is capable of monitoring various motions in a broad working bandwidth. Besides, the peak current exhibits a strong frequency dependence, i.e., the peak current increases from 0.2 to 8.5 nA/cm^2^ with the tested frequency from 1 to 5 Hz. This can be explained as the higher stretching frequency is believed to induce the flow of external electrons in a shorter time, resulting in an enhanced current. By contrast, no output signal can be observed for the pure Ecoflex dielectric elastomer-based strain sensor (Appendix A). The effective instantaneous power of the self-powered sensor, as a vital factor for real-time applications, is characterized by measuring the peak current across different load resistors. A maximum peak power density of 0.18 nW/cm^2^ is achieved with an external load resistance of 8 × 10^8^ Ω, as shown in Figure 2c. Moreover, to ensure its reliability of practical applications, the sensor is cyclically stretched for more than 15,000 cycles with an applied strain. The stable peak current and quantity of transferred charge (ΔQ) indicate excellent stability and durability for this sensor (Figure 2d).

Additionally, the as-fabricated sensor can also be used as a pressure sensor for its compressibility, and the working mechanism is schematically illustrated in Figure 2e. The deformation of the sensor induces the charge/discharge on the electrode by the charged electret. The similar tests are performed to assess the sensing performances of this pressure sensor, and similar results are obtained as with the strain one. With the impacting frequency increasing from 1 Hz to 5 Hz, the peak current is increased from 100 nA/cm^2^ to 200 nA/cm^2^. The peak power density of the sensor reaches 1000 nW/cm^2^ for a load resistance of 10^8^ Ω. The reliability and durability of the pressure sensor are verified by the stable peak current and ΔQ during the 15,000 compression cycles. Compared with the performances in the strain mode, the sensor exhibits intensely enhanced output signals in the pressure mode, which is caused by the triboelectric induction during the contact-separation process [27,28]. Besides, the Ecoflex sealed layer brings about a waterproof feature for the sensor with a water contact angle of 95°. The sensor is immersed in water for 20 h and still maintains a stable output signal, as depicted in Appendix A.

With respect to the feature of diversified current changing outputs corresponding to dual-mode mechanical stimuli, the fabricated sensor is sufficiently applicable to versatile potential applications in wearable electronics, and a demonstration is conducted accordingly in Figure 3. By applying a periodic stretching-releasing motion on the sensor, the collected current signals clearly show an accurate recording of the stretching status, in which the positive current corresponds to the stretching process, while negative current represents the releasing process (Figure 3a). As shown in Figure 3b,c, this device is attached to the finger and wrist joints to detect their movements, and the corresponding changes in current of the sensor according to the applied strain are presented. The bending and straightening of finger or wrist can be clearly distinguished through the positive or negative current changes, and the actions can also be identified according to the magnitude of output signals. When a pressing action is performed (Figure 3d), the difference in current changes is observed according to the motions. A larger peak current is obtained, which is consistent with the results achieved above, as shown in Figure 2f. Moreover, the negligible deviation in output signals under each motion demonstrates the excellent stability and repeatability of this sensor.

## 4. Conclusions

In summary, a long-lived stretchable composite electret, consisting of polarized ePTFE membrane and Ecoflex elastomer, is achieved through a straightforward coating approach. The obtained stretchable electret exhibits a stable surface potential of −0.38 kV over 30 days, large strain of 450%, low tensile stress, low hysteresis, and translucence. It is applicable to assemble a self-powered flexible sensor to detect external stimuli, including pressure and strain. The sensor exhibits excellent stability, durability, fast response, water resistance, and frequency dependence in both pressure and strain modes. Interestingly, the output signals in the pressure mode are almost two orders of magnitude larger than those in the strain mode for the synergistic effect of triboelectric and electrostatic induction. Moreover, the sensing mechanism in each mode is investigated. Eventually, the sensor can be mounted to skin, thereby perceiving and monitoring human motions, paving a path towards the future applications of stretchable electrets in wearable electronic devices.

## Figures and Tables

**Figure 1 nanomaterials-13-00158-f001:**
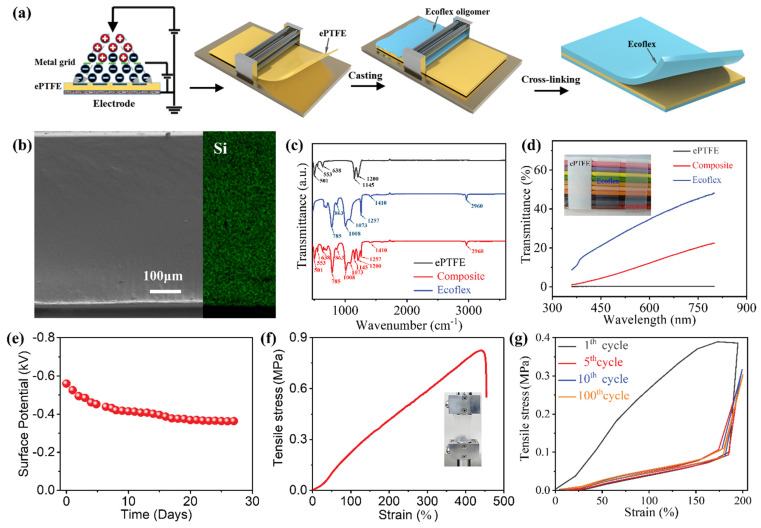
(**a**) Schematic diagram of fabrication process of the stretchable composite electret. (**b**) The cross-sectional SEM image and its corresponding EDX elemental mapping of the composite electret. (**c**) FTIR spectrum of ePTFE, Ecoflex and composite electret. (**d**) Transmittance of the ePTFE, Ecoflex and composite electret in the visible wavelength range. Inset: photograph of the ePTFE, Ecoflex and composite electret placed on color pens. (**e**) Surface potential decay of the composite electret film with negative charges. (**f**) Stress-strain curve of the composite electret under uniaxial tensile test. (**g**) Stress-strain curve of the composite electret under cyclic tensile test (200% strain).

**Figure 2 nanomaterials-13-00158-f002:**
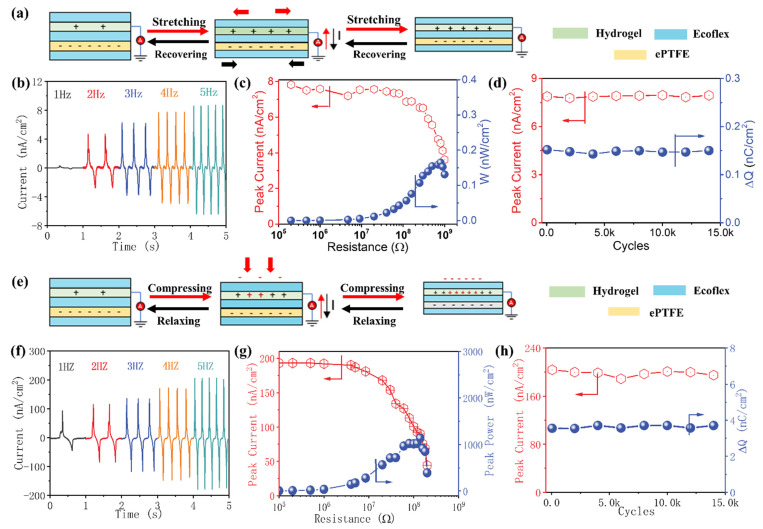
Working mechanisms and output characteristics of the self-powered flexible sensor in the strain (**a**–**d**) and pressure mode (**e**–**h**). (**a**,**e**) Working mechanisms of this self-powered flexible sensor. (**b**,**f**) Time-dependent variation of output current at different vibration frequencies. (**c**,**g**) Output peak current and output power as a function of resistance of external load resistor. (**d**,**h**) Durability tests of the as-fabricated sensor.

**Figure 3 nanomaterials-13-00158-f003:**
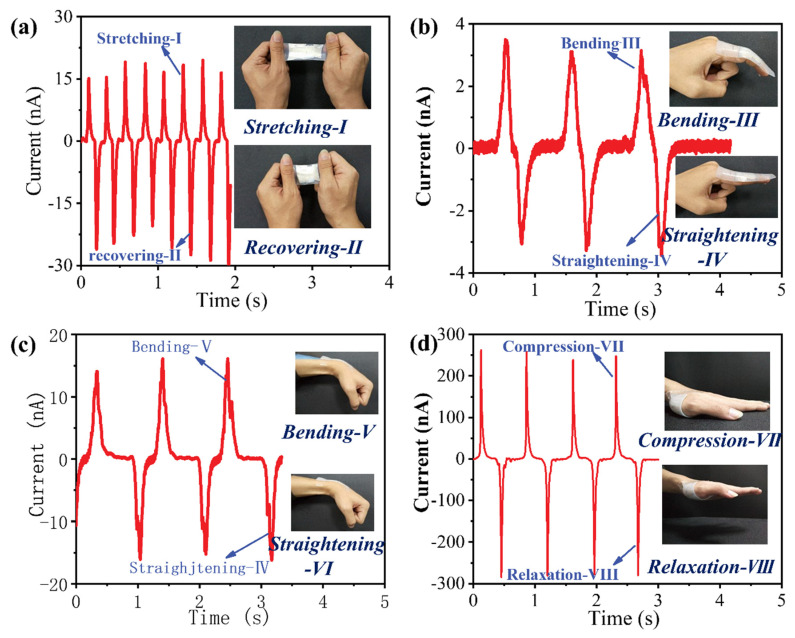
Applications of the flexible sensor for detecting human motions. (**a**) Detected the signals of stretching and releasing motions. (**b**,**c**) Detection of finger bend (**b**) and wrist bend (**c**) by attaching the sensor to the knuckle and wrist joints. (**d**) Real-time monitoring the compression and relaxation actions by mounting the sensor onto the palm.

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
