# Peer review of "A Stretchable Expanded Polytetrafluorethylene-Silicone Elastomer Composite Electret for Wearable Sensor"

_nanomaterials, 2022, doi:10.3390/nano13010158_

Round 1
Reviewer 1 Report
The manuscript "A Stretchable Expanded Polytetrafluorethylene-Silicone Elastomer Composite Electret for Wearable Sensor" reports on the preparation and application of polytetrafluorethylene-Ecoflex composite materials as sensor materials.
Firts of all, I am intrigued by the fact that this paper is not for he purpose of Nanomaterials journal. At most it would fit at Materials journal.
So that, the authors must emphasize more the part of nanomaterials that they develop. The introduction part should thus be improved to emphasize these aspects of nanomaterials.
Secondly, It is not clearly how the both polymers polytetrafluorethylene and Ecoflex were compatibilized to induce such properties. What crosslinking process occurred? The authors must explain that at lines 88-89 and also to highlight in FT-IR spectrum of the composite materials.
What temperature was used? the polymeric precursors were stable at this temperature?
The composite electret was prepared by a hydrogel electrode. It is not clear how and why was this needed. The stability of this sensor in moist media must be studied.
Based on these considerations, I am reserved about the publication of this paper in this journal.
Author Response
Dear Reviewer,
We sincerely thank you for your valuable feedback that would help us to both in English and in depth to improve the quality of our manuscript. Here we submit a new version of our manuscript with the title “A Stretchable Expanded Polytetrafluorethylene-Silicone Elastomer Composite Electret for Wearable Sensor (Manuscript Number: nanomaterials-2082576)”, which has been modified according to your suggestions. Our point-to-point response is given in normal font and changes/additions to the manuscript are given.
The manuscript "A Stretchable Expanded Polytetrafluorethylene-Silicone Elastomer Composite Electret for Wearable Sensor" reports on the preparation and application of polytetrafluorethylene-Ecoflex composite materials as sensor materials.
Firsts of all, I am intrigued by the fact that this paper is not for the purpose of Nanomaterials journal. At most it would fit at Materials journal.
So that, the authors must emphasize more the part of nanomaterials that they develop. The introduction part should thus be improved to emphasize these aspects of nanomaterials.
Secondly, it is not clearly how the both polymers polytetrafluorethylene and Ecoflex were compatibilized to induce such properties. What crosslinking process occurred? The authors must explain that at lines 88-89 and also to highlight in FT-IR spectrum of the composite materials.
What temperature was used? the polymeric precursors were stable at this temperature?
The composite electret was prepared by a hydrogel electrode. It is not clear how and why was this needed. The stability of this sensor in moist media must be studied.
Based on these considerations, I am reserved about the publication of this paper in this journal.
Reply: We feel great thanks for your professional review work on our article. According to your nice suggestions, we have revised the manuscript extensively. And the detailed corrections are listed below.
- The authors must emphasize more the part of nanomaterials that they develop. The introduction part should thus be improved to emphasize these aspects of nanomaterials.
Reply:
Thank you for your kind suggestion. As suggested by you, the development of nanomaterials was supplied in the introduction section.
Scientists have found an alternative way to improve the longevity of elastomer electrets, which is to introduce inorganic nanoparticles or hard plastic nanoparticles, e.g. silica nanoparticles, barium titanate nanoparticles, and PTFE nanoparticles, into the elastomer matrix.
Additionally, in our work, the selected ePTFE membrane possesses some typical characteristics of nanomaterials, e.g. nanoporous and nanofiber structures.
- It is not clearly how the both polymers polytetrafluorethylene and Ecoflex were compatibilized to induce such properties. What crosslinking process occurred? The authors must explain that at lines 88-89 and also to highlight in FT-IR spectrum of the composite materials.
Reply:
Thank you for your kind suggestion. In this work, the interconnected porous network of ePTFE membrane (Fig.S2) and good wettability of Ecoflex oligomer on the ePTFE membrane (Fig.S3) provide a complete infiltration and an outstanding interface interaction between ePTFE nanofibers and silicone rubber. The interconnected silicone rubber network and strong interfacial interaction endow excellent elasticity and stretchability of ePTFE/Ecoflex composites.
During the crosslinking process, a hydrosilylation addition reaction occurred between silicon part A and part B as following:
See Fig.1c, the transmission peaks at 1200 and 1145 cm−1 in the FTIR spectrum of ePTFE corresponded to the antisymmetric and symmetric stretching vibrations of the (CF2) group; the transmission peak at 638 cm−1 corresponded to the bending vibration of the C-F bond; the transmission peak at 553 and 501cm−1 corresponded to the deformation vibration of the C-F bond. The transmission peak at 785 cm-1 in the Ecoflex FTIR spectrum corresponded to the coupling of the stretching vibration of Si-C bond and swaying vibration of -CH3 group; the transmission peak at 1073 and 1008 cm-1 corresponded to the stretching vibration of Si-O bond in the linear Si-O-Si bond; the transmission peak at 1257 and 863 cm-1 corresponded to the bending vibration and swaying vibration of Si-CH3 group; the transmission peak at 2962 cm-1 corresponded to the stretching vibration of -CH3 group; and the transmission peak at 1410 cm-1 corresponded to the shearing vibration of CH=CH2 bond in Si-CH=CH2 group. For the ePTFE/Ecoflex composites, the characteristic transmission peaks are the same as the ePTFE and Ecoflex, indicating that no fresh chemical bond formed between ePTFE and Ecoflex in the course of cross-linking process.
- What temperature was used? the polymeric precursors were stable at this temperature?
Reply:
Thank you for your kind comment. The polymeric precursors were thoroughly mixed and cured at room temperature for 4h and post cured at 80 oC for 2h in this work. The polymeric precursors are stable at this temperature. And more detail information is available at www.smooth-on.com.
- The composite electret was prepared by a hydrogel electrode. It is not clear how and why was this needed. The stability of this sensor in moist media must be studied.
Reply:
Thank you for your kind suggestion. In this paper, the hydrogel electrode was employed for its advantages of stretchability and stable conductivity during the deformation. The stability of this sensor in moist media was studied and the results were presented in Fig.S7.

Reviewer 2 Report
The paper: A Stretchable Expanded Polytetrafluorethylene-Silicone Elastomer Composite Electret for Wearable Sensor, the authors: Jianbo Tan,Kaikai Chen,Jinzhan Cheng,Zhaoqin Song,Jiahui Zhang,Shaodi Zheng*,Shiju E*, Zisheng Xu* , represent an interesting study for Nanomaterials readers, more corrections are necessary.
My principal questions or remarks:
Please write complete the name of author: Shiju E
The title is clear.
The content is in accord with title.
The manuscript adheres to the journal's standards after revision.
The size of the article is appropriate to the contents.
The authors must underline the major findings of their work and explain novelty of this study. The objectives must be better pointed.
The Abstract section refers to the study findings, methodologies, discussion as well as conclusion, but can be completed.
The key words permit found article in the current registers or indexes.
In the introduction is clearly described the state of the art of the investigated problem.
The methods are well described.
Please respect journal format. The Experimental section must be: 2. Materials and Methods… move after 1. Introduction
Please put the reference:
Firstly, the hydrogel electrode was synthesized following the procedure reported by Hyunwoo Yuk et al.
The figures have a good quality. What means each figure must be presented before figures presentation.
The comparison with other articles is necessary.
The Conclusion is OK. The main results are presented in this section.
The paper is easy to understand by readers from other area.
The literature is sufficiently critical, current, and internationally evaluated. Please citation references from last years. It is this study actual?
The paper was written in standard, grammatically correct English, small corrections are necessary.
Please completed:
Supporting Information is available from the ...... or from the author.
Please verify references and respect guide for authors. The journals name must be abbreviated.
In manuscript I don’t find S8 (this figure is in supplementary materials).
Author Response
Dear Reviewer,
We sincerely thank you for your valuable feedback that would help us to both in English and in depth to improve the quality of our manuscript. Here we submit a new version of our manuscript with the title “A Stretchable Expanded Polytetrafluorethylene-Silicone Elastomer Composite Electret for Wearable Sensor (Manuscript Number: nanomaterials-2082576)”, which has been modified according to your suggestions. Our point-to-point response is given in normal font and changes/additions to the manuscript are given.
My principal questions or remarks:
Please write complete the name of author: Shiju E
The title is clear.
The content is in accord with title.
The manuscript adheres to the journal's standards after revision.
The size of the article is appropriate to the contents.
The authors must underline the major findings of their work and explain novelty of this study. The objectives must be better pointed.
The Abstract section refers to the study findings, methodologies, discussion as well as conclusion, but can be completed.
The key words permit found article in the current registers or indexes.
In the introduction is clearly described the state of the art of the investigated problem.
The methods are well described.
Please respect journal format. The Experimental section must be: 2. Materials and Methods… move after 1. Introduction
Please put the reference:
Firstly, the hydrogel electrode was synthesized following the procedure reported by Hyunwoo Yuk et al.
The figures have a good quality. What means each figure must be presented before figures presentation.
The comparison with other articles is necessary.
The Conclusion is OK. The main results are presented in this section.
The paper is easy to understand by readers from other area.
The literature is sufficiently critical, current, and internationally evaluated. Please citation references from last years. It is this study actual?
The paper was written in standard, grammatically correct English, small corrections are necessary.
Please completed:
Supporting Information is available from the ...... or from the author.
Please verify references and respect guide for authors. The journals name must be abbreviated.
In manuscript I don’t find S8 (this figure is in supplementary materials).
Reply:
Thanks for your professional review work on our article. According to your nice suggestions, we have revised the manuscript extensively. And the detailed corrections are listed below.
- Please write complete the name of author: Shiju E
Reply:
Thank you for your kind remind. Shiju E is the complete spelling of the name of author.
- The Abstract section refers to the study findings, methodologies, discussion as well as conclusion, but can be completed.
Reply:
Thank you for your kind comment. As suggested by you, the Abstract section was revised.
- Please respect journal format. The Experimental section must be: 2. Materials and Methods… move after 1. Introduction
Reply:
Thank you for your kind comment. As suggested by you, the format of the article has been revised.
- Please put the reference:
Firstly, the hydrogel electrode was synthesized following the procedure reported by Hyunwoo Yuk et al.
Reply:
Thank you for your kind remind. The literature was cited in ref.23 in the revised manuscript.
- The literature is sufficiently critical, current, and internationally evaluated. Please citation references from last years. It is this study actual?
Reply:
Thank you for your kind comment. We sincerely appreciate the valuable comment, and the corresponding references have been cited appropriately in ref.9, 20, 25, 26.
- Please completed:
Supporting Information is available from the ...... or from the author.
Reply:
Thank you for your kind comment. The following supporting information can be downloaded at: https://www.mdpi.com. Fig.S1. Surface potential decay of the ePTFE membrane and Ecoflex film. Fig.S2. SEM image of the pristine ePTFE membrane. Fig.S3. Contact angles of water (a) and Ecoflex oilgomer (b) on the surface of ePTFE membrane. Fig.S4. Photos displaying superior flexibility of the electret composite under different deformation (a) rolling up, (b) folding, (c) twisting and (d) crumpling. Fig.S5. Stress-strain curves of the ePTFE film (a) and Ecoflex (b) under uniaxial tensile test. (c) Comparison of the tensile strain and stress of the ePTFE, Ecoflex and electret composite. Stress-strain curves of the ePTFE film (d) and Ecoflex (e) under cyclic tensile test (200% strain). Fig.S6. The variation of output current (a) and voltage (b) of Ecoflex and electret composite under periodic tensile test. Fig.S7. (a)The water contact angle of the as-fabricated sensor. (b) The photo of the sensor soaked in the water. (c) The peak current and DQ of the sensor as a function of soak time. Fig.S8. (a) Basic structure diagram of stretchable electret. (b) Basic physical model of stretchable electret.
- Please verify references and respect guide for authors. The journals name must be abbreviated.
Reply:
Thank you for your kind comment. As suggested by you, the journals names were abbreviated.
- In manuscript I don’t find S8 (this figure is in supplementary materials).
Reply:
Thank you for your kind comment. The description of Fig.S8 is replenished in revised manuscript.

Reviewer 3 Report
The article, entitled «A Stretchable Expanded Polytetrafluorethylene-Silicone Elastomer Composite Electret for Wearable Sensor», reports the preparation of non-nanosized all-polymer film by Doctor Blade. The prepared film was further modified with a special hydrogel in order to make a prototype of a wearable sensor. The film itself was characterized with Scanning Electron Microscopy, Charge Decay, Optical and Mechanical measurements. The sensor was able to sense the stretching, bending and pressure resulting in the respective voltage signals. It was proposed to use the sensor for the detection of the human hand movement.
The article makes quite positive impression. Nevertheless, I’d would recommend its resubmission to the «Polymers» by MDPI as the article only discusses the non-nanosized all-polymer film and its application for a wearable sensor.
MINOR:
- Regarding the transmittance measurements, the was no any peak in the graphs. What was actually meant by the «intuitive» comparison? Is the transmittance important for application of the films as sensors?
- The photos of stretching, bending and pressing along with I-V curves are shown in the Fig. 3. It’s mentioned in the text, the aforementioned actions can be distinguished from the I-V curves. However, there was no thorough explanation how to do it automatically. Please provide some comments on that matter.
TYPOS and STYLE:
- Page 1 line 21-22: «commercial inorganic (silicon dioxide (SiO2))[8, 9], barium titanate (BaTiO3)[10]», the inorganic electrets are only represented by silicon dioxide and barium titanate. No need to write the chemical formulas and write the words «commercial inorganic»
- Page 1 lines 21-24: excessive use of abbreviations, e.g. «(PZT)». It doesn’t make any sense to introduce an abbreviation if you only use it once in the text.
- Page 3 line 88: «no fresh absorption peaks», I think that the word «fresh» isn’t suitable for the description of the peaks.
- In the manuscript, there are no spaces and brackets in the figure notations, e.g. «Fig.2d». More common notation is «Fig. 2 (d)»
- Page 6 line 187: «a path toward», instead of «a path towards»
Author Response
Dear Reviewer,
We sincerely thank you for your valuable feedback that would help us to both in English and in depth to improve the quality of our manuscript. Here we submit a new version of our manuscript with the title “A Stretchable Expanded Polytetrafluorethylene-Silicone Elastomer Composite Electret for Wearable Sensor (Manuscript Number: nanomaterials-2082576)”, which has been modified according to your suggestions. Our point-to-point response is given in normal font and changes/additions to the manuscript are given.
Comments and Suggestions for Authors
The article, entitled «A Stretchable Expanded Polytetrafluorethylene-Silicone Elastomer Composite Electret for Wearable Sensor», reports the preparation of non-nanosized all-polymer film by Doctor Blade. The prepared film was further modified with a special hydrogel in order to make a prototype of a wearable sensor. The film itself was characterized with Scanning Electron Microscopy, Charge Decay, Optical and Mechanical measurements. The sensor was able to sense the stretching, bending and pressure resulting in the respective voltage signals. It was proposed to use the sensor for the detection of the human hand movement.
The article makes quite positive impression. Nevertheless, I’d would recommend its resubmission to the «Polymers» by MDPI as the article only discusses the non-nanosized all-polymer film and its application for a wearable sensor.
MINOR:
- Regarding the transmittance measurements, there was no any peak in the graphs. What was actually meant by the intuitive comparison? Is the transmittance important for application of the films as sensors?
Reply: Thank you for your suggestion. Generally, the inorganic or hard polymeric electrets are opaque. With the development of flexible electronic devices, the transparency of the sensor could be another important feature to broaden its applications. In this work, the transmittance of composite electret is enhanced by eliminating the light scattering for the cellular-free and uniform structure, which provides a guidance for the fabrication of the transparent flexible electret-based sensor.
- The photos of stretching, bending and pressing along with I-V curves are shown in the Fig. 3. It’s mentioned in the text, the aforementioned actions can be distinguished from the I-V curves. However, there was no thorough explanation how to do it automatically. Please provide some comments on that matter.
Reply: Thank you for your nice comment. Upon stretching the sensor, the distance between electret film and electrode decreases with the deformation of sensor, an increasing charge on the electrode is induced by the charged electret. In turn, the increasing gap and declining charge is evoked during the recovering process. When a pressure is loaded, the deformation of the sensor induces the charge/discharge on the electrode by the charged electret. Thus, the feature in charge/discharge on the electrode can be applied to distinguish the motions.
TYPOS and STYLE:
- Page 1 line 21-22: «commercial inorganic (silicon dioxide (SiO2))[8, 9], barium titanate (BaTiO3)[10]», the inorganic electrets are only represented by silicon dioxide and barium titanate. No need to write the chemical formulas and write the words commercial inorganic
Reply: Thank you for your nice comment. As suggested by you, the unnecessary chemical formulas were removed.
- Page 1 lines 21-24: excessive use of abbreviations, e.g. (PZT). It doesn’t make any sense to introduce an abbreviation if you only use it once in the text.
Reply: Thank you for your nice comment. As suggested by you, the unnecessary abbreviations were removed.
- Page 3 line 88: «no fresh absorption peaks», I think that the word «fresh» isn’t suitable for the description of the peaks.
Reply: Thank you for your nice comment. As suggested by you, replace the “fresh” with “new”.
- In the manuscript, there are no spaces and brackets in the figure notations, e.g. «Fig.2d». More common notation is «Fig. 2 (d)»
Reply: Thank you for your nice comment. As suggested by you, the figure notations were revised.
- Page 6 line 187: «a path toward», instead of «a path towards»
Reply: Thank you for your nice comment. As suggested by you, the grammatical mistake was revised.

Reviewer 4 Report
The paper contains interesting results about a sensor devices obtained by assembling different polymeric materials with complementary properties. The devices and the work as described relates the interesting ultimate properties of the sensor device with the macrostructure ,whereas molecular7nano features and their role are not discussed . This last part should be added. Some points are indicated below
1.The Abstract should be more descriptive of the work reported in the paper.
the composite electret was solidified at 80 °C for 2 h. t2.The ePTFE and ECOFLEX structure and properties should be reported in detail in the experimental part.
3 lines 79-80 Thermal crosslinking is vague; how this occurs ,? Are additives used to obtain this and what structural changes produces ?
4 .lines 88-90 the ms reports “Fourier transform infrared spectroscopy (Fig.1c), no fresh absorption peak is presented 88 for the composite electret, indicating that no chemical reaction occurred between ePTFE 89 and elastomer in the course of cross-linking process”, but the crosslinking is based on new bonds ( in limited amounts ) formation ? .This sense is not clear
5.The preparation method and the figures 1 and 2 indicate that the sensor is finally obtained by layers of elastomer and gels on both sides of the ePTFE membranes . This last , being the less stretchable should determine the final stretching extent. What is the mechanism by which the elastomer improve s the PTFE strain ? This should means that the PTFE is dispersed in the elastomer matrix ?
6,Lines 194-5 “the composite electret was solidified at 80 °C for 2 h.“ This sense is not clear<, what does solidified means in relation to the composite structure.
T
Author Response
Dear Reviewer,
We sincerely thank you for your valuable feedback that would help us to both in English and in depth to improve the quality of our manuscript. Here we submit a new version of our manuscript with the title “A Stretchable Expanded Polytetrafluorethylene-Silicone Elastomer Composite Electret for Wearable Sensor (Manuscript Number: nanomaterials-2082576)”, which has been modified according to your suggestions. Our point-to-point response is given in normal font and changes/additions to the manuscript are given.
The paper contains interesting results about sensor devices obtained by assembling different polymeric materials with complementary properties. The devices and the work as described relates the interesting ultimate properties of the sensor device with the macrostructure, whereas molecular7nano features and their role are not discussed. This last part should be added. Some points are indicated below
1.The Abstract should be more descriptive of the work reported in the paper.
Reply:
Thank you for your kind comment. As suggested by you, the Abstract section was revised.
Soaring developments in wearable electronics raise an urgent need for stretchable electrets. However, achieving soft electrets simultaneously possessing excellent stretchability, longevity and high charge density is still challenging. Herein, a facile approach is proposed to prepare an all-polymer hybrid composite electret based on the coupling of elastomer and ePTFE membrane. The composite electrets are fabricated via a facile casting and thermal curing process. The obtained soft composite electrets exhibit constantly high surface potential (-0.38 kV) over a long time (30 days), large strain (450%), low hysteresis, and excellent durability (15000 cycles). To demonstrate the applications, the stretchable electret is utilized to assemble a self-powered flexible sensor based on electrostatic induction effect for human activities monitoring. Additionally, the output signals in the pressure mode almost two orders of magnitude larger than those in the strain mode is observed and the sensing mechanism in each mode is investigated.
- The ePTFE and ECOFLEX structure and properties should be reported in detail in the experimental part.
Reply:
Thank you for your kind comment. As suggested by you, the ePTFE and Ecoflex structure and properties were supplied in detail in the experimental part.
- lines 79-80 Thermal crosslinking is vague; how this occurs? Are additives used to obtain this and what structural changes produces?
Reply:
Thank you for your kind comment. During the crosslinking process, a hydrosilylation addition reaction occurred between silicon part A and part B as following:
And no additional additive was used in this work.
4 .lines 88-90 the ms reports “Fourier transform infrared spectroscopy (Fig.1c), no fresh absorption peak is presented 88 for the composite electret, indicating that no chemical reaction occurred between ePTFE 89 and elastomer in the course of cross-linking process”, but the crosslinking is based on new bonds ( in limited amounts ) formation ? This sense is not clear
Reply:
Thank you for your suggestion.
See Fig.1c, the transmission peaks at 1200 and 1145 cm−1 in the FTIR spectrum of ePTFE corresponded to the antisymmetric and symmetric stretching vibrations of the (CF2) group; the transmission peak at 638 cm−1 corresponded to the bending vibration of the C-F bond; the transmission peak at 553 and 501cm−1 corresponded to the deformation vibration of the C-F bond. The transmission peak at 785 cm-1 in the Ecoflex FTIR spectrum corresponded to the coupling of the stretching vibration of Si-C bond and swaying vibration of -CH3 group; the transmission peak at 1073 and 1008 cm-1 corresponded to the stretching vibration of Si-O bond in the linear Si-O-Si bond; the transmission peak at 1257 and 863 cm-1 corresponded to the bending vibration and swaying vibration of Si-CH3 group; the transmission peak at 2962 cm-1 corresponded to the stretching vibration of -CH3 group; and the transmission peak at 1410 cm-1 corresponded to the shearing vibration of CH=CH2 bond in Si-CH=CH2 group. For the ePTFE/Ecoflex composites, the characteristic transmission peaks are the same as the ePTFE and Ecoflex, indicating that no fresh chemical bond formed between ePTFE and Ecoflex in the course of cross-linking process.
5.The preparation method and the figures 1 and 2 indicate that the sensor is finally obtained by layers of elastomer and gels on both sides of the ePTFE membranes. This last, being the less stretchable should determine the final stretching extent. What is the mechanism by which the elastomer improves the PTFE strain? This should mean that the PTFE is dispersed in the elastomer matrix?
Reply:
Thank you for your comment. In this work, the interconnected porous network of ePTFE membrane (Fig.S2) and good wettability of Ecoflex oligomer on the ePTFE membrane (Fig.S3) provide a complete infiltration and an outstanding interface interaction between ePTFE nanofibers and silicone rubber. After curing, the ePTFE nanofibers were thoroughly coated by the elastomer. The interconnected silicone rubber network and strong interfacial interaction endow excellent elasticity and stretchability of ePTFE/Ecoflex composites.
6, Lines 194-5 “the composite electret was solidified at 80 °C for 2 h.“ This sense is not clear<, what does solidified means in relation to the composite structure.
Reply:
Thank you for your comment. I am sorry for our less rigorous description of the experiment to confuse you, and a clearer description is given in revised manuscript.
Finally, the Ecoflex precursor was solidified at room temperature for 4h and post cured at 80 oC for 2h to obtain the composite electret.

Round 2
Reviewer 1 Report
I have read the revised manuscript and the authors' response to the observations made. In Response to the reviewer the authors mentioned that they solved the main issues of this paper, especially those referring to the nanomaterials field. I did not found these information in the revised manuscript, in Introduction section. The other observations have been clarified.
Based on this observation I recommend the revision of this manuscript.
Author Response
Dear Reviewer 1,
We sincerely thank you for your valuable feedback. Here we submit a new version of our manuscript with the title “A Stretchable Expanded Polytetrafluorethylene-Silicone Elastomer Composite Electret for Wearable Sensor (Manuscript Number: nanomaterials-2082576)”, which has been modified according to your suggestions. Our point-to-point response is given in normal font and changes/additions to the manuscript are given.
- I have read the revised manuscript and the authors' response to the observations made. In Response to the reviewer the authors mentioned that they solved the main issues of this paper, especially those referring to the nanomaterials field. I did not find the information in the revised manuscript, in Introduction section. The other observations have been clarified.
Based on this observation I recommend the revision of this manuscript.
Reply:
Thank you for your kind suggestion. As suggested by you, the development of nanomaterials was given in the introduction section as following:
A straightforward approach has been proposed to improve the longevity of elastomer electrets, which is to introduce the inorganic nanoparticles or hard plastic nanoparticles, e.g. silica nanoparticles [9] and PTFE nanoparticles [22], into the elastomer matrix. (Page 2 in revised manuscript)
A thermally polarized ePTFE porous membrane, consisting of interconnected nanopores and nanofibrous PTFE, is introduced into Ecoflex elastic matrix by a facile doctor-blade casting process. (Page 3 in revised manuscript)

Reviewer 2 Report
The manuscript was improved in accord with recommendations.
Author Response
Dear reviewer:
Thanks so much for your kind assessment of our work. On behalf of my co-authors, we would like to express our great appreciation.
Thank you and best regards.
Yours sincerely,
Zisheng Xu
Round 3
Reviewer 1 Report
The authors have improved their manuscript highlighting more the novelty of this work in the field of nanomaterials, so that I recommend the publication of this paper in the present form.